# Mycetoma: A critical appraisal of educational content on YouTube

Ivaan Pitua[1], Raafidha Raizudheen[1], Amelia Margaret Namiiro[1], Lorraine Apili[1], Felix Bongomin [2,3]*

1 Makerere University College of Health Sciences, Kampala, Uganda, 2 Department of Medical Microbiology and Immunology, Faculty of Medicine, Gulu University, Gulu, Uganda, 3 Department of Internal Medicine, Gulu Regional Referral Hospital, Gulu, Uganda

* drbongomin@gmail.com

## Abstract

### Background

Mycetoma is a fungal neglected tropical disease. Accurate dissemination of information is critical in endemic areas. YouTube, a popular platform for health information, hosts numerous videos on mycetoma, but the quality and reliability of these videos remain largely unassessed.

### Methods

We used modified DISCERN and Global Quality Score (GQS) for reliability and quality respectively. Video duration, views, likes, and comments were recorded. Spearman's rank correlation and Mann-Whitney U tests were used to identify correlations between metrices and quality scores.

### Results

We included 73 mycetoma-related YouTube videos were analyzed, the median GQS score was 4.00 ((IQR = 3.33–4.00), indicating generally high-quality content, while the median mDISCERN score was 3.00 (IQR = 3.00–3.00) reflecting moderate reliability. Videos produced by professionals had significantly higher scores compared to those from consumer-generated content (p < 0.001). A significantly positive correlation was observed between video duration and both GQS (r = 0.417, p < 0.001) and mDISCERN (r = 0.343, p = 0.003). However, views, likes and comments did not significantly correlate with video quality. Additionally, videos longer in duration (p < 0.001) and older in upload date (p = 0.014) had higher quality scores.

### Conclusions

The study shows that mycetoma-related videos on YouTube are generally of high quality, with moderate reliability. This emphasizes the need for expert involvement in content creation and efforts to improve health information online.

**Data Availability Statement:** All relevant data are within the manuscript and its supporting information files. Data are freely available on public repository and can be accessed through figshare DOI: 10.6084/m9.figshare.27323598.

**Funding:** The author(s) received no specific funding for this work.

**Competing interests:** The authors have declared that no competing interests exist.

## Author summary

Mycetoma is a neglected tropical disease caused by fungi, and sharing accurate information about it is essential, especially in regions where the disease is common. YouTube, a popular platform for health information, contains many videos related to mycetoma, but their quality and reliability have not been fully assessed. In this study, we analyzed 73 YouTube videos on mycetoma to evaluate their quality and trustworthiness using established scoring methods. We found that most videos were of generally high quality, though their reliability was moderate. Videos created by healthcare professionals were significantly more reliable and of higher quality than those produced by non-professionals. Interestingly, longer videos and older uploads tended to score better. However, viewer engagement, such as the number of views, likes, or comments, did not seem to be linked to video quality. Our findings highlight the importance of expert involvement in producing reliable health content and the need for continued efforts to improve the accuracy of health information online.

## Introduction

Mycetoma is a chronic, progressive infectious disease that destroys the subcutaneous tissues spreading to the skin, deep tissues and bone, is now considered a World Health Organization priority fungal disease of global public health significance [1–3]. Mycetoma was recognized as a neglected tropical disease (NTD) at the 69th World Health Assembly (WHA) in May 2016 [4]. Characterized by a symptomatic triad: tumor, fistulas and grains, mycetoma can evolve into severe deformities and disabilities if not adequately treated [5]. Both bacteria (actinomycetoma) and fungi (eumycetoma) are implicated as causative agents, but the filamentous fungus *Madurella mycetomatis* is the main causative agent globally [6]. Although mycetoma can occur in various geographic locations, it is most prevalent in countries with tropical and subtropical climates, mostly between the latitudes 15˚ south and 30˚ north, known as the mycetoma belt [7]. The belt includes, among others, central Africa (Chad, Ethiopia, Mauritania, Sudan, Senegal, and Somalia) as well as Mexico, India, Venezuela, and Yemen with Sudan and Mexico experiencing some of the highest incidence rates. The mycetoma belt is characterized by a hot, dry climate with a short, heavy rainy season. Eumycetoma is mainly endemic in Africa [7, 8]. Mycetoma is often contracted through the traumatic implantation of pathogens into the skin, a scenario that typically occurs when individuals walk barefoot in endemic areas [9]. As a result, the disease is particularly prevalent among agricultural workers, who are more susceptible to sustaining skin injuries while working outdoors [10].

Given the increasing surveillance and reporting of mycetoma cases coupled with the challenges associated with its diagnosis and treatment, there is a growing need for reliable sources of information to educate both patients and healthcare providers about the disease. In recent years, digital platforms, particularly YouTube, have emerged as popular sources of health information. YouTube, with its vast repository of videos on a wide range of topics, offers an easily accessible platform for individuals seeking health-related information [11, 12]. The visual and auditory nature of video content, combined with the ability to reach a global audience, makes YouTube an attractive option for disseminating health information, particularly in regions where traditional healthcare education resources may be scarce [13]. Unlike peer-reviewed medical literature, which undergoes rigorous evaluation before publication, the content on YouTube is not subject to formal review processes, making it difficult to ensure the

quality and reliability of the information provided. Consequentially, consumers must make their own judgments about how much trust to place in a source and the quality of the information it shares, often being influenced by their level of health and digital literacy, prior knowledge, personal situations, and personal beliefs [14].

The quality of health-related YouTube videos vary widely, with some videos offering well-researched, evidence-based information, while others provide outdated, incomplete, or misleading content [15]. Various tools and criteria have been developed to assess the quality of health-related videos. The modified DISCERN instrument and the Global Quality Scale (GQS) offers assessment of video reliability and quality respectively [16]. While it has the potential to serve as a valuable platform for disseminating health information, no scientific evaluation of the reliability and quality of YouTube content on mycetoma has been conducted, our study aimed at exploring these aspects of YouTube videos on mycetoma.

## Materials and methods

### Search strategy

A comprehensive data retrieval was conducted on YouTube on July 30th, 2024, to identify videos related to Mycetoma OR Actinomycetoma OR Eumycetoma. The search utilized the YouTube Data API with "Mycetoma," "Actinomycetoma," and "Eumycetoma" as the primary keywords. An R script was developed to retrieve 200 video URLs per search term through automated API queries, with up to 50 results per query, and pagination was used to ensure the complete retrieval of the targeted number of videos. The search results were ranked by YouTube's default relevance setting. In total, 600 videos were identified and screened (**Fig 1**). For each video, key details such as video URL, view count, like count, comment count, upload date, and video duration were extracted.

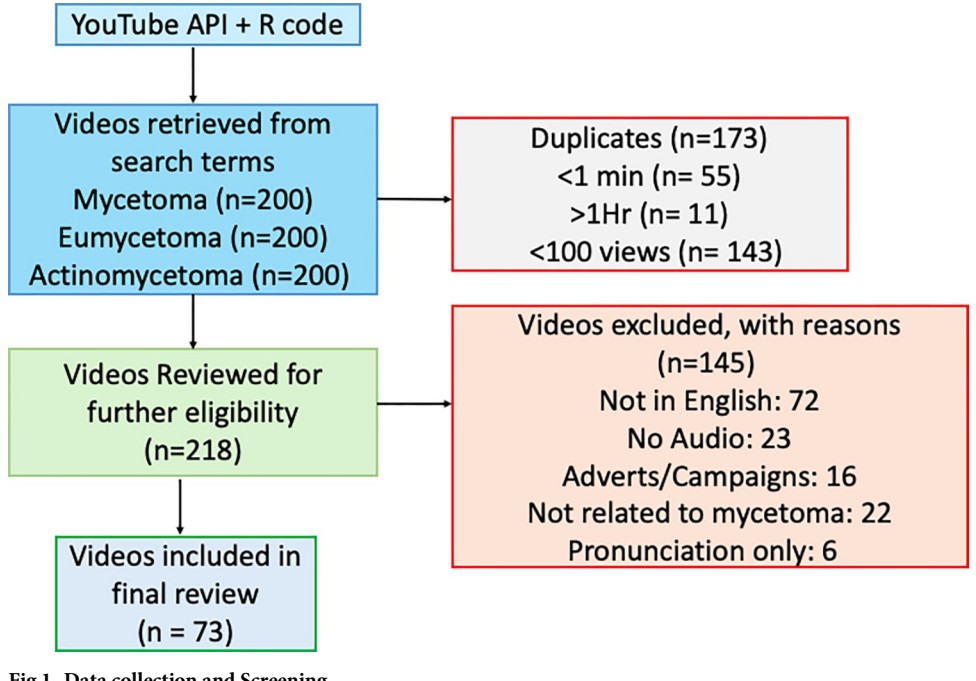

**Fig 1. Data collection and Screening.**

### Eligibility criteria

To be eligible for evaluation, videos had to meet the following inclusion criteria: (1) contain information related to mycetoma; (2) be freely accessible to all users; and (3) be in the English language. Videos were excluded if they were commercials, lacked sound, not related to mycetoma, had pronunciation only, had less than 100 views (Studies on internet search engines have shown >90% of search engine users click on links within the first 3 pages of search results. Most users do not scroll past a few pages of search results (the first 10 YouTube pages usually has 200 videos)[17, 18], when sorted by Relevance, View count or Rating, videos with less than 100 views was found to be falling way outside this range hence exclusion), and had a duration of less than one minute or more than one hour (super-short less than 60 seconds offer insufficient educational content and too long videos greater than 1 hour tend to overload the audience causing mind wandering [19]).

### Data management

Baseline information for all the 600 videos, including URL, length, upload date, number of views, likes was automatically retrieved by the R code into a Microsoft Excel worksheet where the screening was done to come up with 73 videos (S1 Data). Three reviewers (IP, NAM, and LA) independently assessed the videos' quality of information and reliability using GQS and mDISCERN respectively. Prior to the assessment, the reviewers piloted the use of the assessment tools by independently evaluating the first five videos to discuss and resolve any difficulties or questions. Disagreements among raters were noted and resolved by an independent opinion from RR.

### Quality and reliability of information

Reliability was assessed using a 5-point modified DISCERN tool, adapted by Uzun [20] from the original DISCERN tool developed by Charnock et al. [21]. The mDISCERN scale comprises 5 questions each scored either 0 or 1 point with 5 indicating high reliability and a 0-point demonstrates low reliability.

The quality of the videos was evaluated using the 5-point Global Quality Scale (GQS), developed by Bernard and colleagues [22] to assess the quality of information presented on websites. Scores of 1–2 points were considered low quality, 3 points indicated moderate quality, and 4–5 points were considered high quality, **Table 1**.

**Table 1. The modified DISCERN tool and Global Quality Score.**

| mDISCERN (1 point for every Yes, 0 point for every No) | Characteristics for Global Quality Score (GQS) | GQS Score |
|---|---|---|
| Are the aims clear and achieved? | Poor quality, poor flow, most information missing, not useful for patients. | 1 |
| Are reliable sources of information used? | Generally poor, some information given but of limited use to patients. | 2 |
| Are additional sources of information listed for patient reference? | Moderate quality, some important information is adequately discussed. | 3 |
| Are areas of uncertainty mentioned? | Good quality, good flow, most relevant information is covered, useful for patients. | 4 |
| Is the information presented balanced and unbiased? | Excellent quality and excellent flow, very useful for patients. | 5 |

## Statistical analysis

Statistical analyses were conducted using SPSS (IBM SPSS Statistics version 27, IBM Corporation, Armonk, NY, USA). Normal distribution was tested using the Shapiro–Wilk test where applicable. Descriptive statistics, including medians and interquartile ranges, were calculated to summarize the characteristics of the videos. The inter-rater agreement among the three reviewers for the GQS and mDISCERN scores was assessed using the Interclass Correlation Coefficient (ICC). ICC values <0.5 were categorized as indicating poor reliability, values between 0.5 and 0.75 as moderate reliability, values between 0.75 and 0.9 as good reliability, and values >0.90 as excellent reliability. Spearman's rank correlation was used to evaluate the association between video quality scores and video metrics. The Mann-Whitney U test was conducted to compare continuous variables between the "Low to Medium Quality" and "High Quality" groups. Additionally, Chi-Square test was used to assess the relationship between categorical variables, such as the video quality and source of upload (consumer Vs professional, Videos were categorized as consumer-generated if uploaded by individuals, whereas professional content was defined as being uploaded by healthcare institutions, educational bodies, or healthcare workers/experts in the field (These were either mentioned/indicated in video slides or Video titles). This distinction was verified by reviewing the uploader profiles and descriptions as well). Statistical significance was set at $P \leq 0.05$.

## Results

### Screening

We screened 600 YouTube Videos for eligibility, Fig 1.

### Description of variables

Seventy-three videos were included in final analysis, the videos showed a median of 875 views (IQR = 245–2607), 15.5 likes (IQR = 4.00–46.00), and 1 comment (IQR = 0.00–5.00). The median duration of the videos was 8.83 minutes (IQR = 4.02–20.04), with a median of 1372.5 days since upload (IQR = 901–1713). The overall median GQS was 4.00 (IQR = 3.33–4.00) and the median mDISCERN score was 3.00 (IQR = 3.00–3.00). Most of the videos were uploaded in 2020.

### Inter-rater reliability

The ICC for the GQS was 0.872, indicating good reliability, while the ICC for the mDISCERN was 0.805, also suggesting good reliability.

### Correlation analysis

There was a strong positive correlation was observed between mDISCERN and GQS (r = 0.695, p < 0.001), indicating that higher mDISCERN scores were associated with higher GQS scores. The duration of the video was significantly correlated with both mDISCERN (r = 0.343, p = 0.003) and GQS (r = 0.417, p < 0.001). Also, days after upload was significantly correlated with mDISCERN (r = 0.314, p = 0.007) but not GQS. However, no significant correlations were observed between video quality scores and the number of views, likes, or comments (Table 2).

### Comparison of video quality

High-quality videos had significantly higher GQS (Median = 4.00, IQR = 0.33) and mDISCERN scores (Median = 3.00, IQR = 0.00) compared to low-to-medium quality videos (GQS:

**Table 2. Correlation between video parameters and scoring scales.**

| Video Parameters | | mDISCERN | Global Quality Score |
|---|---|---|---|
| Views | r | 0.129 | 0.075 |
| | p | 0.278 | 0.530 |
| Likes | r | 0.113 | 0.150 |
| | p | 0.343 | 0.206 |
| Comments | r | 0.073 | 0.009 |
| | p | 0.558 | 0.941 |
| Duration of Video (Mins) | r | 0.343 | 0.417 |
| | p | **0.003** | **<0.001** |
| Days Since Upload | r | 0.314 | 0.188 |
| | p | **0.007** | 0.111 |

Median = 3.00, IQR = 0.83; mDISCERN: Median = 2.00, IQR = 1), with both p-values < 0.001. There were no significant differences between the groups regarding the number of views, likes, or comments. High-quality videos were significantly longer in duration (Median = 14.15 minutes, IQR = 19.43) and had been uploaded for a longer period (Median = 1464 days, IQR = 757) compared to low-to-medium quality videos (Duration: Median = 3.83 minutes, IQR = 5.47; Days since upload: Median = 897, IQR = 995), with p-values of 0.000 and 0.014, respectively (**Table 3**).

## Video source and quality

There was a significant association between Source of upload and video quality (p = 0.048), with professional groups more likely to produce high-quality videos (n = 34 out of 41) compared to consumer groups (n = 20 out of 32) (**Table 4**).

## Discussion

Our study provides an assessment of the quality and reliability of YouTube videos related to mycetoma. As the prevalence of this chronic, granulomatous skin infection continues to rise, particularly in tropical and subtropical regions attributable to both increased surveillance, awareness and diagnosis as well as actual incidence cases [5], ensuring access to accurate and reliable information is essential for both patients and healthcare providers. To our knowledge, this is the first study to do such assessment on mycetoma related videos on YouTube.

One significant observation from our study is the positive correlation between video duration and quality scores. Longer videos tend to have higher GQS and mDISCERN scores,

**Table 3. Association between Video Quality and continuous variables.**

| | Video Quality, Median (IQR) | | |
|---|---|---|---|
| | **Low to Medium Quality N = 19** | **High Quality N = 54** | **p-value** |
| Global Quality Score | 3.00 (0.83) | 4.00 (0.33) | **<0.001** |
| mDISCERN | 2.00 (1) | 3.00 (0) | **<0.001** |
| Views | 542.00 (1897) | 898.00 (2912) | 0.945 |
| Likes | 13.00 (35) | 16.00 (45) | 0.811 |
| Comments | 2.00 (5) | 1.00 (5) | 0.473 |
| Duration of Video (Mins) | 3.83 (5.47) | 14.15 (19.43) | **<0.001** |
| Days Since Upload | 897.00 (995) | 1464.00 (757) | **0.014** |

**Table 4. Association between Source of upload and video quality.**

| | | Source of Upload | | Total |
|---|---|---|---|---|
| | | Consumer Group | Professional Group | |
| Video Quality | Low to Medium Quality | 12 | 7 | 19 |
| | High Quality | 20 | 34 | 54 |
| Total | | 32 | 41 | 73 |

indicating that more extensive content may provide more comprehensive and higher-quality information. This result supports findings from previous studies that suggest comprehensive videos are more likely to deliver thorough, well-researched content [13]. However, it is crucial to balance video length with viewer engagement to avoid information overload. The short length of the videos is intended to prevent the viewer from becoming too passive and to prevent thoughts from wandering. Moreover, some videos had less than 5% of the entire 1 hour length covering the topic in question which could have been solved by segmenting the videos and uploading them in parts [23]. The other category of video formats that was not retrieved by our code is "Shorts" introduced by YouTube in 2021. These are videos that last less than 60 seconds and usually attract more views and likes per view but less comments per view compared to regular videos. They mostly target entertainment unlike regular videos that cover wide ranges of categories. Shorts do not outperform regular videos in the education and political categories as much as they do in other categories [24].

Additionally, videos produced by professionals were more likely to achieve higher quality scores compared to those created by consumers. This finding emphasizes the importance of expert input in producing educational content and is consistent with other studies showing that professional or institutional videos generally offer higher quality compared to user-generated content [16]. Cuma and colleagues found that the popularity (VPI), reliability (mDIS-CERN) and quality (GQS) of videos uploaded by healthcare professionals were significantly higher compared to non-healthcare providers, more so, all of the high quality videos were of healthcare professionals [25]. A larger systematic review found that Health professional group videos had more reliability and better quality than the non-health professional group [11]. However, it should always be kept in mind that patients may be exposed to low-quality and unreliable videos with misleading content.

Our analysis also revealed that video metrics such as views, likes, and comments do not significantly correlate with video quality scores suggesting that the popularity of a video does not necessarily reflect its accuracy or reliability. Many studies do attest to this finding [26, 27], which must be resolved by encouraging expert groups to provide and continuously promote high quality video content to YouTube users and patients. Viewers should be cautious and critically evaluate content, regardless of its popularity. Despite the overall moderate to high quality of many videos in our study, a significant proportion may not meet the desired standards of accuracy and reliability. This points to the need for ongoing efforts to enhance the quality of health-related content on digital platforms and to educate content creators about the importance of evidence-based information.

While our study did not systematically evaluate the depth of scientific content in the mycetoma-related videos, it was observed that there were notable gaps in the information provided, particularly concerning accurate diagnostic methods and treatment protocols as most sources were not thorough in this aspect. This lack of reliable content raises concerns about misinformation, which can erode trust in medical and public health institutions. The findings of this study have significant implications for public health, particularly in regions where mycetoma is endemic. Accurate information about mycetoma's prevention, diagnosis, and management

is essential to reduce the burden of this neglected tropical disease. Public health campaigns must target not only at-risk populations but also healthcare workers, ensuring they are equipped with the latest knowledge on diagnosing and treating mycetoma. Training and improving knowledge of lower health cadres such as VHTs can improve case detection [28], it is also imperative to integrate community engagement and education to the care of people infected with mycetoma as well as their relatives and the community at large [29]. Integrating online platforms like YouTube into health education efforts offers both challenges and opportunities in disseminating reliable information, underlining the need for collaborations between health authorities and digital platforms to curb misinformation. Healthcare workers are advised to prioritize early diagnosis through clinical assessment and confirmatory diagnostic tests given the complexity of mycetoma and its overlap with other infections.

This study has several limitations that should be considered when interpreting the findings. First, our search strategy focused on the top 200 videos per search term retrieved using specific keywords, ranked by YouTube's relevance algorithm. While this approach reflects the method by which most casual users access information on YouTube, it may have missed less popular but potentially high-quality videos. Additionally, YouTube's metrics, such as views and likes, are dynamic and can change rapidly; therefore, the data presented in this study are accurate only as of the date of the search. Our study was limited to reliability and quality as per the tools used, in depth information on prevention, diagnosis and treatment in each video was not extracted. Despite these limitations, efforts were made to ensure reliability, including the use of multiple independent reviewers and the calculation of inter-rater reliability using ICCs. Future research should explore alternative search strategies, incorporate other validated video-specific assessment tools, do in depth content analysis and consider the inclusion of all video formats to provide a more comprehensive evaluation.

## Conclusions

Mycetoma-related videos on YouTube are generally of high quality, with moderate reliability. This emphasizes the need for expert involvement in content creation and efforts to improve health information online.

## Supporting information

**S1 Data. Raw dataset.**
(XLSX)

## Author Contributions

**Conceptualization:** Ivaan Pitua, Felix Bongomin.

**Data curation:** Ivaan Pitua, Raafidha Raizudheen, Amelia Margaret Namiiro, Lorraine Apili.

**Formal analysis:** Ivaan Pitua, Amelia Margaret Namiiro, Lorraine Apili.

**Funding acquisition:** Ivaan Pitua.

**Investigation:** Ivaan Pitua, Raafidha Raizudheen.

**Methodology:** Ivaan Pitua, Amelia Margaret Namiiro, Lorraine Apili, Felix Bongomin.

**Project administration:** Ivaan Pitua.

**Resources:** Ivaan Pitua.

**Software:** Ivaan Pitua.

**Supervision:** Ivaan Pitua, Felix Bongomin.

**Validation:** Ivaan Pitua.

**Visualization:** Ivaan Pitua.

**Writing – original draft:** Ivaan Pitua, Raafidha Raizudheen, Amelia Margaret Namiiro, Lorraine Apili, Felix Bongomin.

**Writing – review & editing:** Ivaan Pitua, Raafidha Raizudheen, Amelia Margaret Namiiro, Lorraine Apili.

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
