## [Decision Letter · Decision Letter 0]

10 Oct 2024

Dear Dr. Bongomin,

Thank you very much for submitting your manuscript "Mycetoma: Critical Appraisal of Educational Content on YouTube" for consideration at PLOS Neglected Tropical Diseases. As with all papers reviewed by the journal, your manuscript was reviewed by members of the editorial board and by several independent reviewers. In light of the reviews (below this email), we would like to invite the resubmission of a significantly-revised version that takes into account the reviewers' comments. 

We cannot make any decision about publication until we have seen the revised manuscript and your response to the reviewers' comments. Your revised manuscript is also likely to be sent to reviewers for further evaluation.

Sincerely,

Ahmed Hassan Fahal, FRCS, FRCSI, FRCSG, MS, MD, FRCP(London)

Academic Editor

Joshua Nosanchuk

Section Editor

Reviewer's Responses to Questions

**Key Review Criteria Required for Acceptance?**

**Methods**

-Are the objectives of the study clearly articulated with a clear testable hypothesis stated?

-Is the study design appropriate to address the stated objectives?

-Is the population clearly described and appropriate for the hypothesis being tested?

-Is the sample size sufficient to ensure adequate power to address the hypothesis being tested?

-Were correct statistical analysis used to support conclusions?

-Are there concerns about ethical or regulatory requirements being met?

Reviewer #1: 3. Methods: The 3 assessors – are they experienced in the clinical or laboratory aspects of mycetoma? Are they doctors? All presentations need to address particular audiences – so what audience are they?

4. Needs more clarification of DISCERN tool and Global Quality Score – what are the metrics for these results? What is being measured?

Reviewer #2: - It might be helpful to add why videos with less than 100 views were excluded? Is this standard based on past research in video evaluation? 

- Consider including how researchers distinguished between consumer and professional groups who uploaded the videos

Reviewer #3: The methodology is well laid out.

**Results**

-Does the analysis presented match the analysis plan?

-Are the results clearly and completely presented?

-Are the figures (Tables, Images) of sufficient quality for clarity?

Reviewer #1: 5. Results: is there any data or correlation between the source country and quality? The WHO Mycetoma Collaborating centre is unique. Could these videos be regarded as the ‘highest quality benchmark’ against which to judge others?

6. Presumably with a median duration of 8 minutes, it was impossible for the shorter videos to address clinical presentation, diagnosis and therapy. So how did the these categories of video perform on the quality scores?

7. Can the authors make any general comments about what makes a good video for mycetoma? Clinical photographs? Easily readable slides? Demonstration of diagnostic procedure(s)? Pace of presentation? Depth of data? The reader has no real sense of what the scores are really measuring?

8. The paper should have the listing of all videos scored as a supplementary excel file. Others in the future may wish to address this.

Reviewer #2: Consider putting the actual range of the IQR which may be easier to comprehend quickly for the reader. Right now, it seems like only the upper range is included.

Reviewer #3: The results are clear.

Although the results are interesting, more information on the characteristics of the videos is needed. I suggest include, years, country of publication, type of source (media, user-generated content, health professionals and others) and publication of the videos (news, material created by the user, interviews, advertisements and documentaries).

It is also important to consider information related to the mycetoma discussed in the videos. For example, recommendation for healthcare workers, recommendation for all persons over 18 years of age 15, recommendation for persons 60 years of age or older, diagnosis, treatment, etc.

**Conclusions**

-Are the conclusions supported by the data presented?

-Are the limitations of analysis clearly described?

-Do the authors discuss how these data can be helpful to advance our understanding of the topic under study?

-Is public health relevance addressed?

Reviewer #1: Lack of clarity about defines quality

Reviewer #2: - As in introduction, would consider putting some caveats in about actual rise in prevalence of mycetoma rather than just increased reporting due to awareness and diagnostic access.

- Would move the reasoning for excluding videos based on length into methodology section. 

- While not systemically evaluated, it could be interesting to touch upon the scientific content that was included (diagnosing, symptoms, etc) and the gaps the authors found in terms of actual content. 

- Might be interesting to include some discussion about digital misinformation and the impact on populations trust in medical and public health institutions and how this could impact mycetoma with false information on youtube.

Reviewer #3: Discussion. Include a paragraph with the implications of your findings for public health.

**Editorial and Data Presentation Modifications?**

Reviewer #1: Comments

1. Ref 1 is not a good general reference. There are other reviews, including Ref 3.

2. Ref 3 and 4 about the epidemiology and ‘Mycetoma belt ignore the most recent paper of global distribution: https://doi.org/10.1371/journal.pntd.0008397

Reviewer #2: N/A

Reviewer #3: None

**Summary and General Comments**

Reviewer #1: This is an interesting topic that has not previously seen much research. YouTube as an educational tool in this field has not been analysed, so the study is novel. The study is well written and results are easy to understand.

Reviewer #2: Thank you so much for the opportunity to review this article examining the quality of mycetoma content on Youtube. The authors take a unique approach to evaluating public health and clinical content on mycetoma, and this article could be really interesting in the context of medical and public health disinformation. 

A few considerations for the authors are below: 

- Line 51: It is difficult to state the the prevalence of mycetoma is increasing rather than just being reported more often due to raised awareness and access to diagnostics. With very few countries having surveillance systems, consider modifying this language to reflect that cases are being reported more often.

Reviewer #3: In this study aimed at exploring these aspects of YouTube videos on mycetoma. Findings show mycetoma-related videos on YouTube are generally of high quality, with moderate reliability. 

Although the study design and results are well described. Information for prevention, diagnosis, management etc. is needed. These recommendations are important for public health and tropical diseases. 

Comments: 

Although the results are interesting, more information on the characteristics of the videos is needed. I suggest include, years, country of publication, type of source (media, user-generated content, health professionals and others) and publication of the videos (news, material created by the user, interviews, advertisements and documentaries).

It is also important to consider information related to the mycetoma discussed in the videos. For example, recommendation for healthcare workers, recommendation for all persons over 18 years of age 15, recommendation for persons 60 years of age or older, diagnosis, treatment, etc.

PLOS authors have the option to publish the peer review history of their article (what does this mean?). If published, this will include your full peer review and any attached files.

Reviewer #1: No

Reviewer #2: Yes: Dallas J. Smith

Reviewer #3: No
---

## [Editor Report · Decision Letter 1]

28 Oct 2024

Dear Dr. Bongomin,

We are pleased to inform you that your manuscript 'Mycetoma: A Critical Appraisal of Educational Content on YouTube' has been provisionally accepted for publication in PLOS Neglected Tropical Diseases.

Best regards,

Ahmed Hassan Fahal, FRCS, FRCSI, FRCSG, MS, MD, FRCP(London)

Academic Editor

Joshua Nosanchuk

Section Editor

---

## [Editor Report · Acceptance letter]

7 Nov 2024

Dear Dr. Bongomin,

We are delighted to inform you that your manuscript, "Mycetoma: A Critical Appraisal of Educational Content on YouTube," has been formally accepted for publication in PLOS Neglected Tropical Diseases.

Best regards,

Shaden Kamhawi

co-Editor-in-Chief

Paul Brindley

co-Editor-in-Chief
